# Targeting macrophage Histone deacetylase 3 stabilizes atherosclerotic lesions

Marten A Hoeksema[1], Marion JJ Gijbels[1,2,3], Jan Van den Bossche[1], Saskia van der Velden[1], Ayestha Sijm[1], Annette E Neele[1], Tom Seijkens[1], J Lauran Stöger[1], Svenja Meiler[1], Marieke CS Boshuizen[1], Geesje M Dallinga-Thie[4], Johannes HM Levels[4], Louis Boon[5], Shannon E Mullican[6], Nathanael J Spann[7], Jack P Cleutjens[2], Chris K Glass[7], Mitchell A Lazar[6], Carlie JM de Vries[1], Erik AL Biessen[2], Mat JAP Daemen[8], Esther Lutgens[1,9] & Menno PJ de Winther[1,*]

## Abstract

Macrophages are key immune cells found in atherosclerotic plaques and critically shape atherosclerotic disease development. Targeting the functional repertoire of macrophages may hold novel approaches for future atherosclerosis management. Here, we describe a previously unrecognized role of the epigenomic enzyme Histone deacetylase 3 (Hdac3) in regulating the atherosclerotic phenotype of macrophages. Using conditional knockout mice, we found that myeloid Hdac3 deficiency promotes collagen deposition in atherosclerotic lesions and thus induces a stable plaque phenotype. Also, macrophages presented a switch to anti-inflammatory wound healing characteristics and showed improved lipid handling. The pro-fibrotic phenotype was directly linked to epigenetic regulation of the *Tgfb1* locus upon Hdac3 deletion, driving smooth muscle cells to increased collagen production. Moreover, in humans, *HDAC3* was the sole Hdac upregulated in ruptured atherosclerotic lesions, Hdac3 associated with inflammatory macrophages, and *HDAC3* expression inversely correlated with pro-fibrotic *TGFB1* expression. Collectively, we show that targeting the macrophage epigenome can improve atherosclerosis outcome and we identify Hdac3 as a potential novel therapeutic target in cardiovascular disease.

**Keywords** atherosclerosis; epigenetics; fibrosis; lipids; macrophages
**Subject Categories** Immunology; Vascular Biology & Angiogenesis

## Introduction

Atherosclerosis is a lipid-driven chronic inflammatory disease and the main cause of cardiovascular morbidity and mortality. Macrophages are the most prevalent immune cells in atherosclerotic plaques and play a central role in disease progression (Moore *et al*, 2013). The abundance of macrophages in atherosclerotic plaques, often as lipid-laden foam cells, associates with key features of plaque instability and detrimental clinical outcome (Silvestre-Roig *et al*, 2014). Since current medication and intervention strategies reduce the risk for cardiovascular events only moderately, better understanding of the regulation of inflammatory mechanisms driving disease is essential for further development of new therapeutic approaches.

It is increasingly clear that epigenetic mechanisms govern many aspects of inflammatory and immune responses. Epigenetic enzymes regulate histone tail modifications, such as acetylation and methylation that alter chromatin accessibility and thereby control transcriptional responses (Arrowsmith *et al*, 2012). Epigenetic remodeling of histones regulates macrophages and interventions in histone modifying enzymes in macrophages strongly affect their inflammatory repertoire (reviewed in: Hoeksema *et al*, 2012; Ivashkiv, 2013; Tabas & Glass, 2013). Particularly, Histone deacetylases (Hdacs) regulating the acetylation status of histones are essential in innate immune responses. Broad spectrum Hdac inhibitors are well-known anti-inflammatory agents and reduce inflammation and disease severity in animal models for arthritis, inflammatory bowel disease, and septic shock (Shakespear *et al*, 2011). However, to overcome often-observed unwanted side effects by pan-Hdac inhibition, current efforts in drug development focus on selectively

1  Department of Medical Biochemistry, Experimental Vascular Biology, Academic Medical Center, University of Amsterdam, Amsterdam, The Netherlands
2  Department of Pathology, Maastricht University, Maastricht, The Netherlands
3  Department of Molecular Genetics, Maastricht University, Maastricht, The Netherlands
4  Department of Vascular and Experimental Vascular Medicine, Academic Medical Center, University of Amsterdam, Amsterdam, The Netherlands
5  Bioceros BV, Utrecht, The Netherlands
6  Division of Endocrinology, Diabetes and Metabolism, Department of Medicine, Institute for Diabetes, Obesity and Metabolism, Perelman School of Medicine, University of Pennsylvania, Philadelphia, USA
7  Department of Cellular and Molecular Medicine, University of California, San Diego, CA, USA
8  Department of Pathology, Academic Medical Center, University of Amsterdam, Amsterdam, The Netherlands
9  Institute for Cardiovascular Prevention (IPEK), Ludwig Maximilian's University, Munich, Germany
   *Corresponding author. Tel: +31 20 5666762; E-mail: m.dewinther@amc.uva.nl

targeting of individual Hdacs to improve applicability in treatment of disease (Arrowsmith *et al*, 2012).

We are investigating whether targeting of individual Hdacs is an approach to skew macrophages to a phenotype that dampens atherosclerosis development. Recent data show that Hdac3 is essential for induction of a pro-inflammatory gene program in response to lipopolysaccharide (Chen *et al*, 2012) and that Hdac3 deletion renders macrophages more susceptible to alternative activation (Mullican *et al*, 2011). Here, we present Hdac3 as being critical for regulating the fibrotic phenotype of macrophages. Deletion of Hdac3 in macrophages shifted their phenotype to an atherosclerosis beneficial phenotype leading to enhanced collagen deposition and a stable plaque phenotype. In human, atherosclerosis Hdac3 was associated with plaque vulnerability and negatively correlated with pro-fibrotic *TGFB1* expression.

## Results and Discussion

### Myeloid deletion of Hdac3 enhances collagen deposition in atherosclerotic lesions

We set out to study the role of macrophage Hdac3 in atherosclerotic plaque development by use of a genetic approach. We transplanted atherosclerosis susceptible LDLR$^{-/-}$ mice with bone marrow from either Hdac3$^{fl/fl}$ (Hdac3$^{wt}$) or Hdac3$^{fl/fl}$-LysMCre (Hdac3$^{del}$) mice and subsequently fed them a high cholesterol diet (HCD) for 10 weeks. Upon sacrifice, Hdac3$^{del}$-transplanted mice displayed significantly larger lesions compared to controls (Fig 1A and B). Plaque phenotype analysis showed a major increase in the proportion of collagen rich fibrous cap atheromas and a concomitant decrease in thin fibrous cap atheromas in Hdac3$^{del}$-transplanted mice (Fig 1C). By quantifying the collagen content, we indeed observed increased collagen deposition in atherosclerotic plaques of Hdac3$^{del}$ mice (Fig 1D and E). Polarization microscopy revealed that lesions of Hdac3$^{del}$-transplanted mice had an increase of the most mature and stable red collagen (Junqueira *et al*, 1979) subtype (Supplementary Fig S1A and B). Moreover, a two-fold thicker fibrous cap was observed in the Hdac3$^{del}$-transplanted mice (Fig 1F and G). There was no difference in the amount of α-smooth muscle actin (α-SMA)-positive cells (myofibroblasts and vascular smooth muscle cells, VSMCs)

in the lesions (Fig 1H), indicating that enhanced collagen deposition is not due to increased VSMC numbers. Overall, Hdac3$^{del}$ mice had a favorable plaque phenotype since it showed major features of improved stability and increased plaque size were solely attributable to enhanced collagen deposition.

### VSMCs produce more collagen as a result of enhanced TGF-β secretion by Hdac3$^{del}$ macrophages

To establish the molecular mechanism underlying increased collagen deposition in lesions of Hdac3$^{del}$-transplanted mice, we measured collagen deposition in primary mouse VSMCs treated with supernatants from oxLDL-stimulated Hdac3$^{wt}$ or Hdac3$^{del}$ bone marrow-derived macrophages (BMMs). Interestingly, we observed increased collagen production by VSMCs in response to supernatants from oxLDL-stimulated Hdac3$^{del}$ BMMs (Fig 1I) without effects on VSMC proliferation (Supplementary Fig S2A). These data indicate that Hdac3 deletion induces a soluble macrophage-secreted factor that drives collagen production by VSMCs. To identify candidates mediating such effects, we analyzed microarray data from Hdac3$^{wt}$ and Hdac3$^{del}$ macrophages (Mullican *et al*, 2011). Ingenuity pathway analysis (IPA) revealed that Transforming Growth Factor β (TGF-β) signaling is prominently upregulated in Hdac3$^{del}$ macrophages (Fig 1J). TGF-β is a pleiotropic cytokine that can be secreted by many different cell types, including macrophages and regulatory T cells. It deactivates effector macrophages, inhibits CD4$^+$ T-cell proliferation, and induces a fibrotic program in fibroblasts and smooth muscle cells (Li *et al*, 2006). As a consequence, TGF-β is generally considered to be an anti-atherosclerotic cytokine dampening immune responses and stabilizing atherosclerotic lesions (Mallat *et al*, 2001; Lutgens *et al*, 2002; Reifenberg *et al*, 2012). Further analyzing macrophages, we indeed found enhanced *Tgfb1* gene expression in oxLDL-treated Hdac3$^{del}$ BMMs (Fig 1K) and increased TGF-β secretion by these cells (Fig 1L). Enhanced VSMC collagen production in response to supernatants from Hdac3$^{del}$ BMMs was solely dependent on TGF-β secretion by these cells, as antibody mediated blockade of TGF-β completely abolished the difference in collagen production induced by Hdac3 deletion (Fig 1M). Subsequent analysis of ChIP-seq data (Mullican *et al*, 2011) revealed that Hdac3 directly binds near the *Tgfb1* promoter (Supplementary Fig S2B), and in line with its deacetylase activity, we observed increased histone acetylation at the *Tgfb1* locus in

**Figure 1.    Myeloid Hdac3 deletion results in more stable atherosclerotic lesions.**

A–H    LDLR$^{-/-}$ mice were transplanted with bone marrow from Hdac3$^{wt}$ and Hdac3$^{del}$ donors and sacrificed after 10 weeks HCD. (A, B) Aortic root lesions were stained with Toluidin Blue, and aortic lesion area (*n* = 19/18) was quantified. *P* = 0.0085. (C) Plaque severity (*n* = 19/18) was scored. (D, E) Collagen content was assessed using Sirius Red staining (*n* = 19/18). *P* = 0.0218. (F, G) Minimal cap thickness (*n* = 18/16) was measured at the thinnest region of the fibrotic cap. *P* = 0.003. (H) Vascular smooth muscle cells (VSMC)/myofibroblast area (*n* = 14/13) was quantified using an α-SMA antibody.

I    Primary mouse VSMCs were incubated with bone marrow-derived macrophage (BMM) supernatants of 50 μg/ml oxLDL-stimulated Hdac3$^{wt}$ and Hdac3$^{del}$ BMMs for 24 h. VSMC collagen production was measured and expressed relative to the response to supernatants of Hdac3$^{wt}$ BMMs. One out of four representative experiments is shown. *P* = 0.0001, compared to Hdac3$^{wt}$ macrophages.

J    Upstream regulator analysis was performed on wild-type and Hdac3$^{del}$ BMMs.

K    *Tgfb1* gene expression (*P* = 0.0106) in oxLDL-stimulated BMMs was measured and normalized to household genes.

L    TGF-β secretion (*P* = 0.0057) was measured in the BMM supernatants by ELISA.

M    VSMC collagen was measured after 24 h incubation with BMM supernatants with 20 μg/ml control IgG (*P* = 0.0005) or anti-TGF-β antibody.

N    ChIP for AcH3K9/14 was performed on oxLDL-stimulated BMMs, followed by PCR for the *Tgfb1* gene (*P* = 0.0047) and for the *Igk* gene as a negative control.

Data information: unpaired *t*-test (A–H, K, L), 1-way (I), and 2-way ANOVA (M, N) with Bonferroni correction were performed for statistical analysis. All *in vitro* experiments were performed at least twice in triplicate. Error bars indicate SEM.

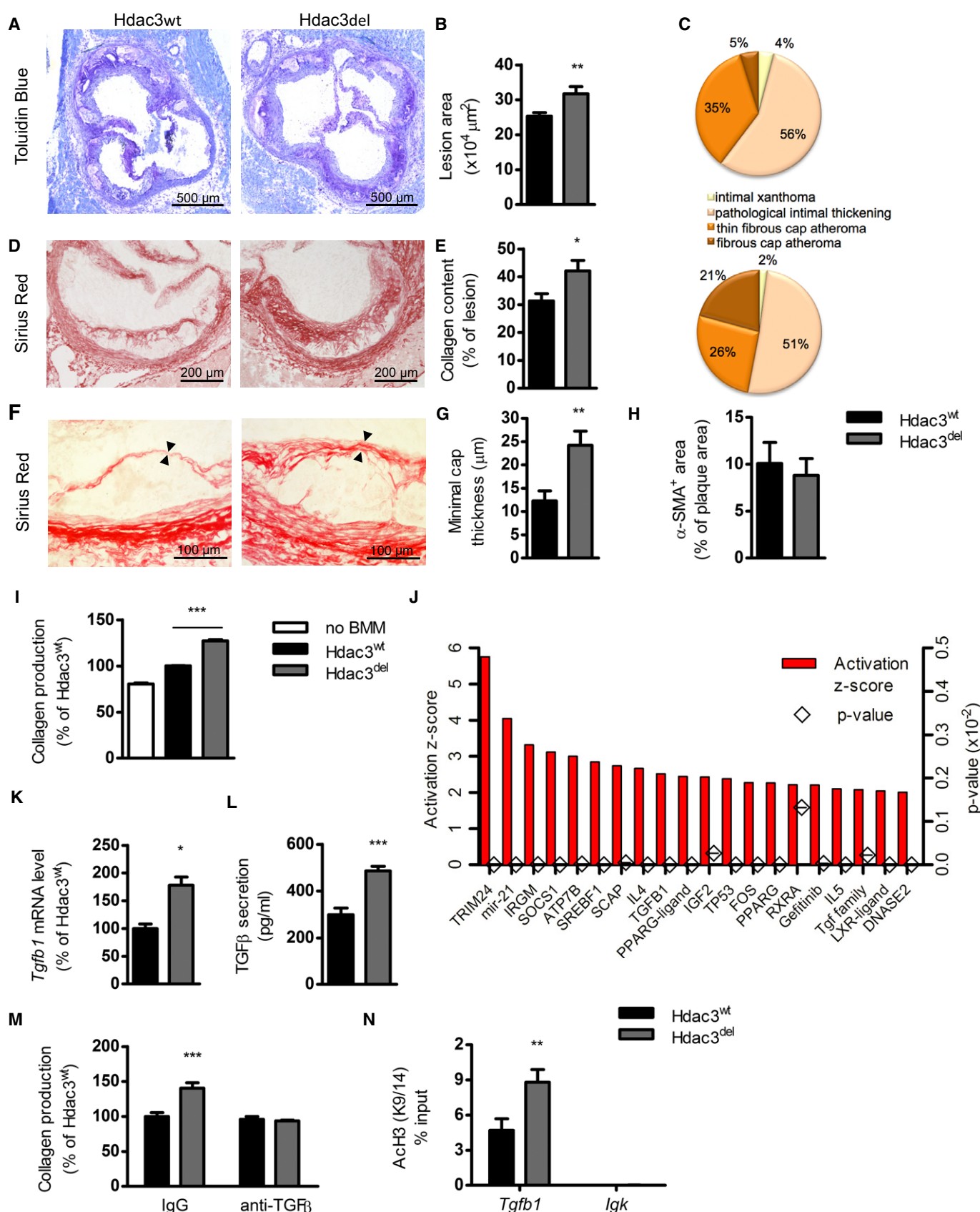

**Figure 1.**

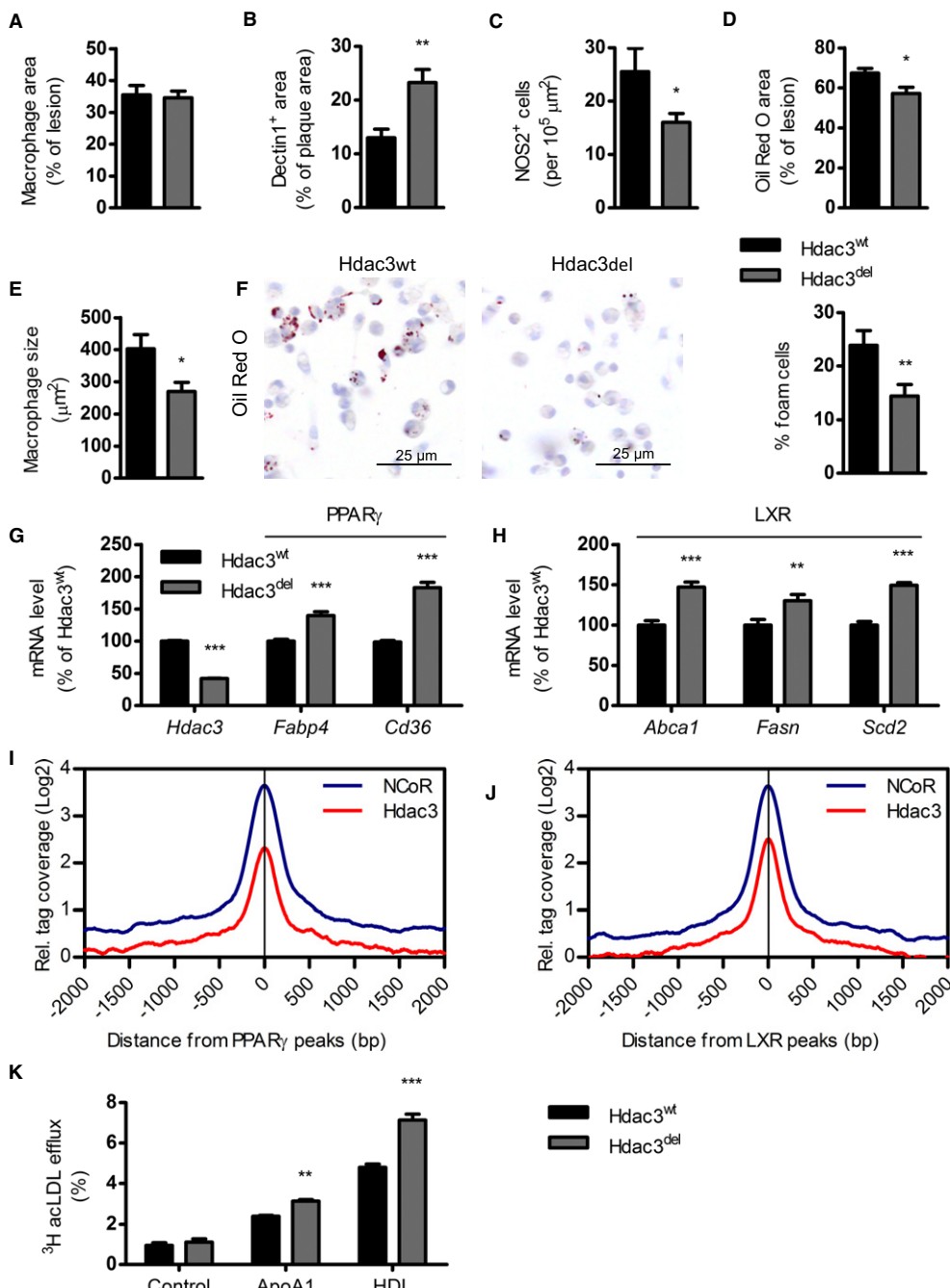

**Figure 2.  Myeloid Hdac3 deletion shifts plaque macrophages phenotype and reduces lipid accumulation.**

A    Macrophage area was determined using Moma-2 staining ($n = 18/17$).

B    Dectin1 ($n = 19/15$, $P = 0.0011$) was expressed as percentage of total lesion area.

C    NOS2[+] ($n = 18/18$, $P = 0.0489$) cells were counted and corrected for lesion size.

D    Lipid content ($n = 19/18$, $P = 0.0149$) was quantified using Oil Red O.

E    Macrophage size within plaques was determined by dividing the size of an allocated macrophage area by the number of macrophages ($n = 10/8$, $P = 0.0238$).

F    PMs from Hdac3[wt] and Hdac3[del]-transplanted LDLR[−/−] mice ($n = 4/5$, $P = 0.0096$) were harvested and stained with Oil Red O, and positive cells were counted as percentage of total macrophages.

G    Gene expression ($n = 4$) of PPARγ target genes, $P = 0.0001$ for all three.

H    Gene expression ($n = 4$) of LXR target genes *Abca1* ($P = 0.0001$), *Fasn* ($P = 0.0018$), and *Scd2* ($P = 0.0001$).

I, J    Distribution of Hdac3 and NCoR ChIP-seq tag densities at PPARγ and LXR peaks.

K    Efflux of AcLDL loaded BMMs to ApoA1 ($P = 0.0074$) or HDL ($P = 0.0001$), $n = 3$.

Data information: unpaired *t*-test (A–F) or, in case of multiple testing, 2-way ANOVA with Bonferroni correction (G, H, K) was performed for statistics. Error bars indicate SEM.

oxLDL-stimulated Hdac3[del] macrophages (Fig 1N). Similarly, Hdac3[del] peritoneal macrophages (PMs) from transplanted LDLR[−/−] mice also showed upregulation of *Tgfb1* expression and increased histone acetylation in the same region (Supplementary Fig S2C–E). Thus, Hdac3 targets the *Tgfb1* locus, and its deletion leads to hyper-acetylation, subsequent increased *Tgfb1* expression and secretion, stimulating VSMCs to produce collagen.

### Hdac3 deletion shifts plaque macrophages to an anti-inflammatory phenotype and reduces lipid accumulation

To further characterize the inflammatory profile of the lesions, immunohistochemical analysis of the plaques was performed. Although total macrophage area was unaltered between groups (Fig 2A), myeloid Hdac3 deletion did result in increased levels of Dectin1[+] macrophages in the lesions (Fig 2B). Interestingly, Dectin1 is a marker for alternatively activated macrophages, and particularly upregulated in wound healing, tissue repair, and fibrosis in which TGF-β is an important mediator (Daley *et al*, 2010; Brancato & Albina, 2011). Concomitantly, there were less NOS2[+] cells (Fig 2C), further implying a decreased pro-inflammatory profile of Hdac3[del] lesions. There was no difference in necrotic core size or immune cell infiltration in the lesions (Supplementary Fig S3). Weight, plasma triglycerides, cholesterol levels, cholesterol profile, and main immune cell populations in the blood were also unaltered (Supplementary Fig S4).

Plaque lipid content (Fig 2D) and macrophage size (Fig 2E) were decreased in lesions of Hdac3[del]-transplanted mice, indicating reduced foam cell formation. Indeed, peritoneal macrophages from Hdac3[del]-transplanted mice revealed decreased lipid accumulation (Fig 2F). Pathway analysis (Fig 1J) indicated that both PPARγ and LXR pathways may be upregulated in Hdac3[del] macrophages. Hdac3 functions in the NCoR repressor complex repressing PPARγ and LXR responses in the absence of ligands (Glass & Ogawa, 2006). Recent data show that deletion of NCoR, disrupting the complex, leads to hyperactivity of LXR and PPARγ in macrophages (Li *et al*, 2013) or adipocytes (Li *et al*, 2011), respectively. Likewise, in the absence of Hdac3, we found several PPARγ and LXR target genes to be upregulated in macrophages (Fig 2G and H). Analysis of ChIP-seq data showed a large overlap between NCoR and Hdac3 peaks and an overlap of approximately 50% of PPARγ and LXR peaks with Hdac3 peaks (i.e., at the *Cd36* locus, Supplementary Fig S5). Consistently, Hdac3 and NCoR tag densities were enriched at both PPARγ and LXR peaks (Fig 2I and J). In line with augmented PPARγ and LXR signaling, Hdac3[del] macrophages showed enhanced lipid efflux capacity (Fig 2K). Overall, myeloid Hdac3 deletion results in less vulnerable lesions that are characterized by reduced lipid accumulation and a shift in macrophage phenotype toward an anti-inflammatory pro-fibrotic gene program.

### HDAC3 is overexpressed in human ruptured atherosclerotic lesions and negatively correlates with plaque-stabilizing TGFB1

Our experimental data in mice show that macrophage Hdac3 deletion shifts atherosclerosis to a more stable, less inflammatory profile indicating a beneficial macrophage phenotype. To evaluate the relevance of these findings for human atherosclerosis, we next studied human atherosclerotic plaque specimen. We first assessed

gene expression of *HDAC1-10* in a cohort of 40 stable versus ruptured human atherosclerotic plaques (Goossens *et al*, 2010). Interestingly, *HDAC3* was the sole Hdac upregulated in ruptured plaques (Fig 3A). Moreover, further analysis of *HDAC3* gene expression in atherosclerotic lesions showed that *HDAC3* strongly correlated with macrophage marker *CD68* (Fig 3B). Immunohisto-chemical analysis showed that Hdac3 particularly localized to CD68[+] rupture-prone shoulder and cap regions of human plaques but not to adventitial areas (Fig 3C and D) indicating a distribution pattern of Hdac3 similar to pro-inflammatory macrophages (Stöger *et al*, 2012). This was further substantiated with the finding that *HDAC3* expression correlated positively with *CD86* (Fig 3E) and *HLA-DPB1* (Supplementary Fig S6) mRNA expression, both markers for activated macrophages. Finally, we found an inverse correlation of *HDAC3* with *TGFB1* gene expression (Fig 3F), corroborating our finding that in mice Hdac3 deletion induces TGF-β expression and a stable plaque phenotype. These data show that Hdac3 is upregulated in ruptured human atherosclerotic plaques, associates with activated macrophages, and inversely correlates with pro-fibrotic *TGFB1* expression.

In conclusion, to our knowledge we are the first to show that targeting the macrophage epigenome can be applied to switch macrophages to an atherosclerosis-favorable phenotype. Our studies show that macrophage Hdac3 deletion is beneficial in atherosclerotic mice. Previous work showed that endothelial silencing of Hdac3 reduces cell survival and enhances atherosclerosis development (Zampetaki *et al*, 2010). These data and our current study thus underscore the cell-specific functions of Hdac3 in atherosclerosis. We found that macrophage-specific targeting increased fibrosis in athero-sclerotic plaques and identified TGF-β as the critical mediator of the pro-fibrotic phenotype of Hdac3[del] macrophages. Moreover, we could demonstrate decreased lipid accumulation, which is also favorable in atherosclerosis. Also in human lesions, Hdac3 was directly linked to plaque stability, as Hdac3 was the sole Hdac upregulated in ruptured lesions and inversely correlated with plaque-stabilizing TGF-β. Our findings show that fine-tuning the macrophage phenotype by altering the epigenetic landscape can be used to influence disease outcome.

## Materials and Methods

### Mice experiments

All mice were on a C57Bl/6 background. LDLR[−/−] mice were obtained from Charles River. Bone marrow transplantation (BMT) was performed as described elsewhere (Goossens *et al*, 2010). Briefly, 40 (20 per group, number based on previous BMT experiments) 10 week old LDLR[−/−] mice were randomly divided over filter-top cages and provided antibiotics water (autoclaved tap water with neomycin (100 mg/l, Sigma) and polymyxin B sulfate (60,000 U/l, Invitrogen)) from 1 week pre-BMT till 5 weeks post-BMT. The animals received $2 \times 6$ Gy total body irradiation on two consecutive days. Bone marrow was isolated from 6 LysMCre-Hdac3[fl/fl] mice (Hdac3[del]) and 6 Hdac3[fl/fl] littermates (Hdac3[wt]), resuspended in RPMI1640 (Gibco) 5 U/ml heparin 2% iFCS (Gibco), and $10^7$ bone marrow cells were injected intravenously per irradiated mouse. Bone marrow transplantation efficiency was determined with qPCR for LDLR on DNA isolated from blood (GE Healthcare),

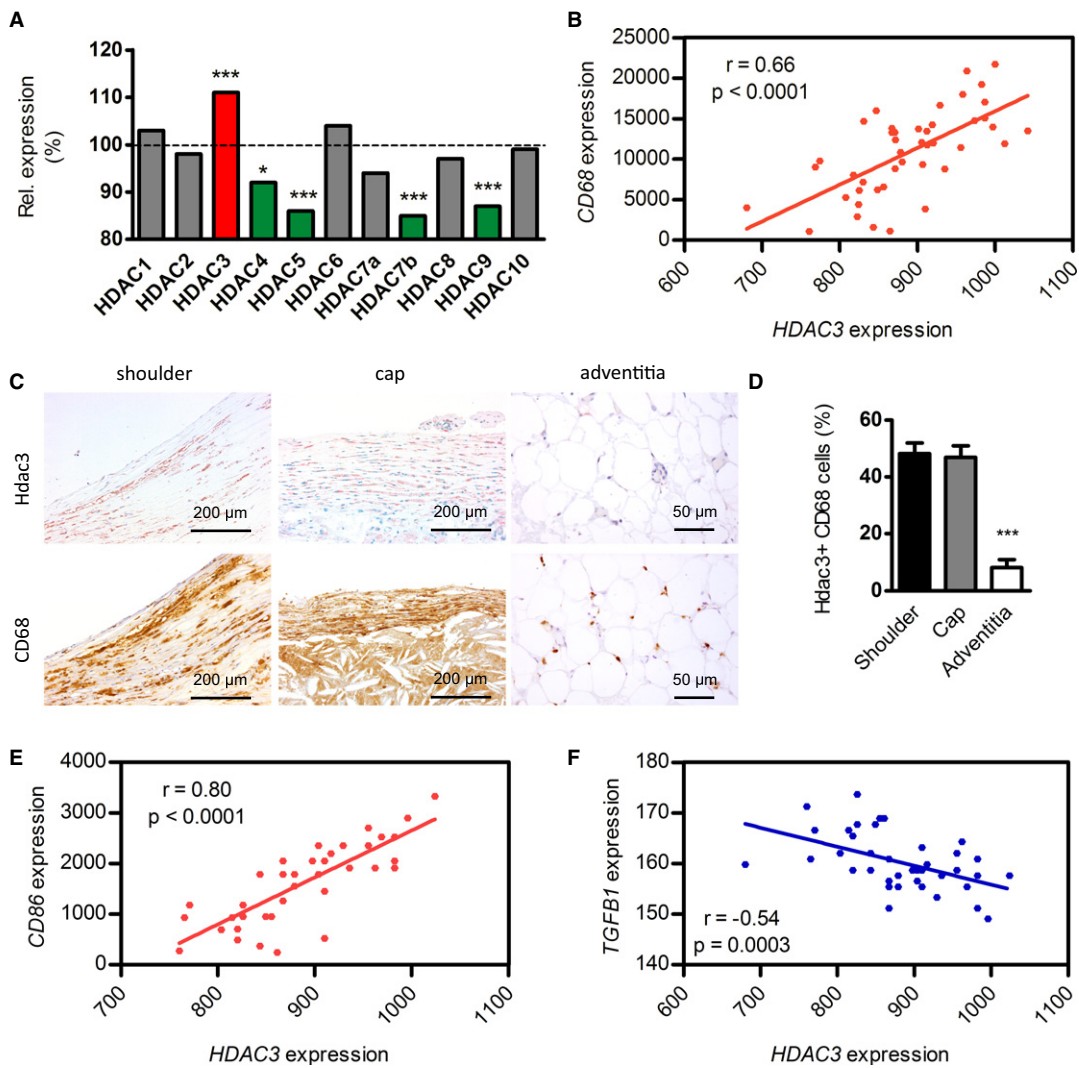

**Figure 3. Hdac3 is expressed in human atherosclerotic plaque macrophages, especially in inflammatory regions.**

A    Microarray which was performed on RNA from ruptured and paired stable control segments (*n* = 20, respectively). *HDAC3* (*P* = 0.00007), *HDAC5* (*P* = 0.00005), *HDAC7b* (*P* = 0.0003), *HDAC9* (*P* = 0.00004).

B    Correlation of *HDAC3* with macrophage marker *CD68* in gene expression data from human atherosclerotic lesions (*n* = 40). Statistical analysis was performed with Pearson's correlation test.

C, D   Hdac3 is present in CD68[+] areas in plaque shoulder (*n* = 21) and cap (*n* = 14) regions, while CD68[+] areas in the adventitia (*n* = 12) showed negligible co-localization with Hdac3. Error bars indicate SEM. *P* = 0.0001 compared to both shoulder and cap regions with 1-way ANOVA.

E    Correlation of *HDAC3* with the pro-inflammatory marker *CD86*.

F    Correlation of *HDAC3* with the pro-fibrotic gene *TGFB1*.

and BMT efficiency was 95.6 ± 0.9% and did not differ between groups. Six weeks after the BMT, the mice were put on a HCD (0.15% cholesterol, 16% fat, Arie Blok Diets) for 10 weeks. After sacrifice, hearts were taken out and frozen in tissue-tek (DAKO) for histology. Blood samples were taken before the start of the diet and before sacrifice for lipid profiling and immune cell FACS and counts. One mouse was sacrificed before the end of experiment as a result of too much weight loss (> 15%), and no other mice were excluded. All animal experiments were conducted at the University of Amsterdam and approved (permit: DBC10AC) by the Committee for Animal Welfare of the Academic Medical Center, University of Amsterdam.

**Histology**

Atherosclerotic lesions from the heart were cut in sections of 7 μm on a Leica 3050 cryostat at −25°C and stained with Toluidin Blue (0.2% in PBS, Sigma) to determine lesion size and Sirius Red (0.05% in saturated picric acid solution, Sigma) for measuring fibrosis. Polarization microscopy on Sirius Red stained slides was performed for identifying collagen subtypes. Positive areas were quantified using Adobe Photoshop 5.5 (Adobe) blindly, without knowledge of genotype. The lesions were also typed blindly according to severity as early (intimal xanthoma), moderate (pathological intimal thickening), and advanced (thin fibrous and

fibrous atheroma), as described before (Kanters et al, 2003; Lutgens et al, 2010). Lesions were fixed in acetone and incubated with antibodies against macrophages (MOMA-2, 1:4,000, AbD serotec), granulocytes (NIMP, directed against Ly6G (Lopez et al, 1984), 1:400, kind gift of P. Heeringa, UMC Groningen), T cells (KT3, directed against CD3, 1:250, AbD serotec), infiltrating macrophages (ER-MP58, 1:100, AbD serotec), and Dectin1 (a-Dectin1, clone BD6, 1:250, AbD serotec). A rabbit anti-rat biotin labeled antibody (1:300, DAKO) was used as secondary antibody. Staining was amplified with an ABC kit and visualized using an AEC kit (both Vector Labs), and cell nuclei were stained with hematoxylin (Merck). For a-SMA (FITC, 1:2,000, Sigma) positive cells, anti-fluorescein-POD (1:500, Roche) was used and visualized using an AEC kit. For NOS2 (1:200, ab15323, Abcam) staining, Brightvision poly-HRP was used as secondary antibody and visualized using an AEC kit and counted blindly. Human lesion stainings were performed as described elsewhere (Stöger et al, 2012). All proceedings were in agreement with the Dutch Code of Conduct for Observational Research with Personal Data (2004) and Tissue (2001). For Hdac3, slides were incubated with a-Hdac3 (1:100, Santa Cruz), and Brightvision poly-HRP was used as secondary antibody and visualized using an AEC kit.

## Lipid profile

Cholesterol and triglyceride levels in the blood were measured with a standard protocol from commercially available kits (Roche). Lipoprotein profiles were determined by fast-performance liquid chromatography (FPLC), as described previously (Seijkens et al, 2014). Individual mouse samples were combined in three pools per genotype, and the data presented are the mean of those three pools.

## Immune cell counts

Blood was collected at sacrifice. Erythrocytes were removed by incubation with erylysis buffer (155 mM $NH_4Cl$, 10 mM $KHCO_3$, 0.1 mM EDTA in PBS, pH 7.4). Non-specific antibody binding was prevented by pre-incubation with an Fcγ-receptor blocking antibody (clone 93, eBioscience). Leukocytes were labeled with CD3-FITC (eBioscience), B220-V500 (eBioscience), CD11b-PeCy7 (BD), Ly6G-PE (BD), and Ly6C-APC (Miltenyi Biotec). Cells were analyzed on a FACSCanto II flowcytometer (BD). The following populations were detected: T cells ($CD3^+$), B cells ($B220^+$), granulocytes ($Ly6G^+$, $CD11b^+$), monocytes ($Ly6G^-$, $CD11b^+$), and $Ly6C^{hi}$ and $Ly6C^{low}$ monocytes.

## Peritoneal macrophages

Bone marrow transplantation was performed, as described. Mice were put on a HCD for 10 weeks and subsequently injected with thioglycollate medium (3%, Fischer). Four days after injection, mice were sacrificed and the peritoneum was flushed twice with 10 ml PBS to collect peritoneal macrophages. Flushed thioglycollate-elicited macrophages were cultured in RPMI-1640 25 mM HEPES, 2 mM L-glutamine, 10% FCS, penicillin (100 U/ml), and streptomycin (100 μg/ml) (all Gibco) and allowed to adhere for 3 h. Oil Red O staining (0.3% in 60% isopropanol, Sigma) was performed to determine lipid accumulation. For gene expression experiments, macrophages adhered overnight.

## Chromatin immunoprecipitation

ChIP was performed as described elsewhere (Spann et al, 2012). ChIP-qPCR was performed on an ABI ViiA7 using SYBR Green Fast (ABI). Relative enrichments are presented as percentage input. Primer sequences are available on request.

## ChIP sequencing and microarray pathway analysis

Data analysis was performed using HOMER (Heinz et al, 2010) on previously published ChIP experiments: GSE33596 (Mullican et al, 2011), GSE27060 (Barish et al, 2012), GSE21314 (Lefterova et al, 2010), and GSE50944 (Li et al, 2013). Each sequencing experiment was normalized to a total of $10^7$ uniquely mapped tags and visualized by preparing custom tracks for the UCSC genome browser. Microarray data were available through GSE33596, and pathway analysis was performed using IPA. Upstream regulator analysis was performed, and upstream regulators were ranked on z-score, which is a statistical test used to define the overlap between differentially expressed genes and genes that are known to be affected by the upstream regulator.

## BMM culture

Femurs and tibia were flushed with ice-cold PBS. Bone marrow cells were cultured in RPMI-1640 25 mM HEPES, 2 mM L-glutamine, 10% FCS, penicillin (100 U/ml), and streptomycin (100 μg/ml) (all Gibco) supplemented with 15% L929-conditioned medium for 8 days. BMMs were stimulated with 50 μg/ml oxLDL (BTI) for 24 h. qPCR analysis was performed for gene expression, and supernatants were used for VSMC collagen production assays.

## Cholesterol efflux

Macrophages were loaded overnight with 25 μg/ml $^3$H-labeled acLDL in RPMI-1640 with 1% BSA. Cells were washed four times with PBS-2% BSA. Then, they were incubated with RPMI-1640 with and without ApoA1 and HDL as acceptors. After 4 h, macrophages were lysed in 2-propanol and radioactivity was measured in both media and cell lysates.

## RNA analysis

RNA was isolated with High Pure RNA Isolation kits (Roche) from 500,000 cells according to manufacturer's protocol. cDNA was synthesized from 500 ng total RNA with iScript (Bio-rad). qPCR was performed with 4 ng cDNA using Sybr Green Fast (Applied Biosystems) on a ViiA7 PCR machine (Applied Biosystems). Gene expression was normalized to the mean of ARBP and GAPDH expression. Primer sequences are available on request.

## VSMC culture

Primary mouse VSMCs were isolated and cultured in DMEM/F12 20% FCS (Gibco) for a maximum of 10 passages on 0.1% gelatin coated plates (Rensing et al, 2014). Fifty thousand VSMCs were plated per well; after overnight adherence, cells were starved (0% FCS) for 48 h and then incubated with macrophage supernatants for

24 h with and without 20 µg/ml control IgG or anti-TGF-β antibody (Dasch *et al*, 1989). For collagen production measurement, cells were fixed in 3.7% formaldehyde, stained with 1% Sirius Red in 0.01 M HCl, and lysed with 0.01 M NaOH, as previously described (Rensing *et al*, 2014). Absorption was measured on a standard plate reader at 544 nm, and a gelatin standard was used for quantification. VSMC proliferation was measured using Celltrace CSFE Cell Proliferation Kit (10 µM, Invitrogen) according to manufacturer's instructions.

### Human transcriptomics

Gene expression data were taken from an existing database and analyzed as described previously (Goossens *et al*, 2010; Stöger *et al*, 2012). Briefly, microarray analysis was performed on RNA from ruptured and paired stable control segments (*n* = 20, respectively) from human endarterectomy specimens that were obtained from the Maastricht Pathology Tissue Collection. Plaque progression stage was assessed by examination of hematoxylin–eosin stained slides from flanking segments using the Virmani classification criteria by a cardiovascular pathologist and an experienced researcher in cardio-vascular pathology. Segments designated as stable featured either a fibrous cap atheroma or pathological intimal thickening. Segments designated as ruptured included a thrombus and/or presented intraplaque hemorrhage. Sections were considered stable or ruptured if they were flanked at both sides within the same endar-terectomy specimen, by segments of identical progression stage. Only endarterectomy specimens were used, which contained both stable and ruptured plaque segments. All use of tissue and patient data was in agreement with the "Code for Proper Secondary Use of Human Tissue in the Netherlands". Illumina Human Sentrix-8 V2.0 BeadChip technology was used to determine mRNA expression.

### Statistical analysis

Data are presented as mean $\pm$ SEM. The statistical analyses were performed using GraphPad Prism 5.0 software using unpaired *t*-test and 1-way or 2-way ANOVA with Bonferroni correction for grouped analyses. *P*-values < 0.05 were considered statistically significant.

**Supplementary information** for this article is available online: http://embomolmed.embopress.org

### Acknowledgements
The authors thank Alinda Schimmel and Linda Beckers for technical assistance, Mariska Vos for VSMC isolation and Noam Zelcer for critically reading the manuscript. J.V.d.B. received a Junior Postdoc grant from the Netherlands Heart Foundation (NHF; 2013T003). Work in the lab of M.A.L. was funded by NIH R01 DK43806. E.L. is an established investigator of the NHF (2009T034), VICI reci-pient (ZonMW, 016130676), holds an AMC fellowship, and is part of the Neth-erlands CardioVascular Research Initiative (CVON2011-19) and SFB1123 (Project A5; Deutsche Forschungs Gemeinschaft (DFG)). M.P.J.d.W is an Estab-lished Investigator of the NHF (2007T067), supported by NHF (2010B022), NWO (TOP91208001), the Netherlands CardioVascular Research Initiative (CVON2011-19) and holds an AMC fellowship.

### Author contributions
MAH designed and carried out all experiments, analyzed the data, and wrote the manuscript. MJJG acted as blinded pathologist by analyzing atherosclerosis severity, all stainings and IHC on lesions. JVdB co-wrote the manuscript and together with AS and AEN helped with performing mouse and *in vitro* experi-ments. SvdV performed animal studies and IHC. TS and SM helped with mouse experiments and performed FACS experiments. JLS and MCSB helped with mouse experiments. GMD and JHML helped with lipid profile and efflux experiments. SEM and MAL provided datasets and bone marrow for BMT and macrophage cultures and acted as advisors. LB provided critical reagents. JPC helped with polarization microscopy. EALB and MJAPD analyzed human atherosclerotic lesion expression data and provided human atherosclerotic lesions for IHC. Other co-authors NJS, CKG, CJdV, and EL acted as advisors, provided critical reagents and reviewed the manuscript. MPJdW supervised the project and wrote the manuscript.

### Conflict of interest
The authors declare that they have no conflict of interest.

### For more information
Microarray data and ChIP-seq data are available through GSE33596.

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

---

**The paper explained**

**Problem**

Atherosclerosis is a lipid-driven chronic inflammatory disease and the main cause of cardiovascular morbidity and mortality. Present medi-cation and interventional therapies reduce the risk for cardiovascular disease only moderately, and recurrent events happen in a significant proportion of patients. Inflammation is a hallmark of atherosclerosis, and the abundance of inflammatory cells, and more specifically macrophages, associates with key features of plaque instability and clinical outcome. We are studying epigenetic approaches to skew macrophages to a phenotype that dampens atherosclerosis develop-ment.

**Results**

Here, we identify the epigenetic enzyme Hdac3 as an important modulator of the fibrotic phenotype of macrophages in atheroscle-rosis. Deletion of Hdac3 stabilized atherosclerotic plaques, skewed the macrophage phenotype to anti-inflammatory characteristics, and improved their lipid handling. Also in humans, we could link the expression of Hdac3 to plaque vulnerability to rupture.

**Impact**

Our findings identify Hdac3 as a potential novel therapeutic target in cardiovascular disease and demonstrate that fine-tuning the macro-phage phenotype by altering the epigenetic landscape can influence disease outcome.

Chen X, Barozzi I, Termanini A, Prosperini E, Recchiuti A, Dalli J, Mietton F, Matteoli G, Hiebert S, Natoli G (2012) Requirement for the histone deacetylase Hdac3 for the inflammatory gene expression program in macrophages. *Proc Natl Acad Sci U S A* 109: E2865–E2874

Daley JM, Brancato SK, Thomay AA, Reichner JS, Albina JE (2010) The phenotype of murine wound macrophages. *J Leukoc Biol* 87: 59–67

Dasch JR, Pace DR, Waegell W, Inenaga D, Ellingsworth L (1989) Monoclonal antibodies recognizing transforming growth factor-beta. Bioactivity neutralization and transforming growth factor beta 2 affinity purification. *J Immunol* 142: 1536–1541

Glass CK, Ogawa S (2006) Combinatorial roles of nuclear receptors in inflammation and immunity. *Nat Rev Immunol* 6: 44–55

Goossens P, Gijbels MJ, Zernecke A, Eijgelaar W, Vergouwe MN, van der Made I, Vanderlocht J, Beckers L, Buurman WA, Daemen MJ *et al* (2010) Myeloid type I interferon signaling promotes atherosclerosis by stimulating macrophage recruitment to lesions. *Cell Metab* 12: 142–153

Heinz S, Benner C, Spann N, Bertolino E, Lin YC, Laslo P, Cheng JX, Murre C, Singh H, Glass CK (2010) Simple combinations of lineage-determining transcription factors prime cis-regulatory elements required for macrophage and B cell identities. *Mol Cell* 38: 576–589

Hoeksema MA, Stoger JL, de Winther MP (2012) Molecular pathways regulating macrophage polarization: implications for atherosclerosis. *Curr Atheroscler Rep* 14: 254–263

Ivashkiv LB (2013) Epigenetic regulation of macrophage polarization and function. *Trends Immunol* 34: 216–223

Junqueira LC, Bignolas G, Brentani RR (1979) Picrosirius staining plus polarization microscopy, a specific method for collagen detection in tissue sections. *Histochem J* 11: 447–455

Kanters E, Pasparakis M, Gijbels MJ, Vergouwe MN, Partouns-Hendriks I, Fijneman RJ, Clausen BE, Forster I, Kockx MM, Rajewsky K *et al* (2003) Inhibition of NF-kappaB activation in macrophages increases atherosclerosis in LDL receptor-deficient mice. *J Clin Invest* 112: 1176–1185

Lefterova MI, Steger DJ, Zhuo D, Qatanani M, Mullican SE, Tuteja G, Manduchi E, Grant GR, Lazar MA (2010) Cell-specific determinants of peroxisome proliferator-activated receptor gamma function in adipocytes and macrophages. *Mol Cell Biol* 30: 2078–2089

Li MO, Wan YY, Sanjabi S, Robertson AK, Flavell RA (2006) Transforming growth factor-beta regulation of immune responses. *Annu Rev Immunol* 24: 99–146

Li P, Fan W, Xu J, Lu M, Yamamoto H, Auwerx J, Sears DD, Talukdar S, Oh D, Chen A *et al* (2011) Adipocyte NCoR knockout decreases PPARgamma phosphorylation and enhances PPARgamma activity and insulin sensitivity. *Cell* 147: 815–826

Li P, Spann NJ, Kaikkonen MU, Lu M, da Oh Y, Fox JN, Bandyopadhyay G, Talukdar S, Xu J, Lagakos WS *et al* (2013) NCoR repression of LXRs restricts macrophage biosynthesis of insulin-sensitizing omega 3 fatty acids. *Cell* 155: 200–214

Lopez AF, Strath M, Sanderson CJ (1984) Differentiation antigens on mouse eosinophils and neutrophils identified by monoclonal antibodies. *Br J Haematol* 57: 489–494

Lutgens E, Gijbels M, Smook M, Heeringa P, Gotwals P, Koteliansky VE, Daemen MJ (2002) Transforming growth factor-beta mediates balance between inflammation and fibrosis during plaque progression. *Arterioscler Thromb Vasc Biol* 22: 975–982

Lutgens E, Lievens D, Beckers L, Wijnands E, Soehnlein O, Zernecke A, Seijkens T, Engel D, Cleutjens J, Keller AM *et al* (2010) Deficient CD40-TRAF6 signaling in leukocytes prevents atherosclerosis by skewing the immune response toward an antiinflammatory profile. *J Exp Med* 207: 391–404

Mallat Z, Gojova A, Marchiol-Fournigault C, Esposito B, Kamate C, Merval R, Fradelizi D, Tedgui A (2001) Inhibition of transforming growth factor-beta signaling accelerates atherosclerosis and induces an unstable plaque phenotype in mice. *Circ Res* 89: 930–934

Moore KJ, Sheedy FJ, Fisher EA (2013) Macrophages in atherosclerosis: a dynamic balance. *Nat Rev Immunol* 13: 709–721

Mullican SE, Gaddis CA, Alenghat T, Nair MG, Giacomin PR, Everett LJ, Feng D, Steger DJ, Schug J, Artis D *et al* (2011) Histone deacetylase 3 is an epigenomic brake in macrophage alternative activation. *Genes Dev* 25: 2480–2488

Reifenberg K, Cheng F, Orning C, Crain J, Kupper I, Wiese E, Protschka M, Blessing M, Lackner KJ, Torzewski M (2012) Overexpression of TGF-beta1 in macrophages reduces and stabilizes atherosclerotic plaques in ApoE-deficient mice. *PLoS ONE* 7: e40990

Rensing KL, de Jager SC, Stroes ES, Vos M, Twickler MT, Dallinga-Thie GM, de Vries CJ, Kuiper J, Bot I, von der Thusen JH (2014) Akt2/LDLr double knockout mice display impaired glucose tolerance and develop more complex atherosclerotic plaques than LDLr knockout mice. *Cardiovasc Res* 101: 277–287

Seijkens T, Hoeksema MA, Beckers L, Smeets E, Meiler S, Levels J, Tjwa M, de Winther MP, Lutgens E (2014) Hypercholesterolemia-induced priming of hematopoietic stem and progenitor cells aggravates atherosclerosis. *FASEB J* 28: 2202–2213

Shakespear MR, Halili MA, Irvine KM, Fairlie DP, Sweet MJ (2011) Histone deacetylases as regulators of inflammation and immunity. *Trends Immunol* 32: 335–343

Silvestre-Roig C, de Winther MP, Weber C, Daemen MJ, Lutgens E, Soehnlein O (2014) Atherosclerotic plaque destabilization: mechanisms, models, and therapeutic strategies. *Circ Res* 114: 214–226

Spann NJ, Garmire LX, McDonald JG, Myers DS, Milne SB, Shibata N, Reichart D, Fox JN, Shaked I, Heudobler D *et al* (2012) Regulated accumulation of desmosterol integrates macrophage lipid metabolism and inflammatory responses. *Cell* 151: 138–152

Stöger JL, Gijbels MJ, van der Velden S, Manca M, van der Loos CM, Biessen EA, Daemen MJ, Lutgens E, de Winther MP (2012) Distribution of macrophage polarization markers in human atherosclerosis. *Atherosclerosis* 225: 461–468

Tabas I, Glass CK (2013) Anti-inflammatory therapy in chronic disease: challenges and opportunities. *Science* 339: 166–172

Zampetaki A, Zeng L, Margariti A, Xiao Q, Li H, Zhang Z, Pepe AE, Wang G, Habi O, de Falco E *et al* (2010) Histone deacetylase 3 is critical in endothelial survival and atherosclerosis development in response to disturbed flow. *Circulation* 121: 132–142

