## [Review Process File · EMBO Molecular Medicine]

Targeting macrophage Histone deacetylase 3 stabilizes atherosclerotic lesions

Marten A. Hoeksema, Marion J. J. Gijbels, Jan Van den Bossche, Saskia van der Velden, Ayestha Sijm, Annette E. Neele, Tom Seijkens, J. Laurant Stöger, Svenja Meiler, Marieke C.S. Boshuizen, Geesje M. Dallinga-Thie, Johannes H.M. Levels, Louis Boon, Shannon E. Mullican, Nathanael J. Spann, Jack P. Cleutjens, Chris K. Glass, Mitchell A. Lazar, Carlie J. de Vries, Erik A.L. Biessen, Mat J.A.P. Daemen, Esther Lutgens, Menno P.J. de Winther

Corresponding author: Menno de Winther, University of Amsterdam

Review timeline:	Submission date:	14 April 2014
	Editorial Decision:	07 May 2014
	Revision received:	02 June 2014
	Editorial Decision:	05 June 2014
	Revision received:	06 June 2014
	Accepted:	10 June 2011

Transaction Report:

Editor: Roberto Buccione

1st Editorial Decision

07 May 2014

Thank you for the submission of your Report manuscript to EMBO Molecular Medicine. We have now received comments from the three Reviewers whom we asked to evaluate your manuscript

You will see that all three Reviewers are quite supportive of your work, although they do raise a few issues that prevent us from considering publication at this time. I will not dwell into much detail, as the evaluations are self-explanatory.

Reviewer 1 has various requests for clarification on results and experimental details and would also like to see, if available, data from the entire aorta.

Reviewer 2 is more critical in that s/he notes that the study is limited to the "prophylactic" setting and would like to see if adoptive transfer of Hdac3^{-/-} monocytes/macros into LDLR^{-/-} with established lesions also has an effect so as to give the manuscript a "curative" angle as well. Such experiments would substantially increase the impact of the study but even if not required for publication, without such data the authors should be required to temper this conclusion. Although I will not be requiring you to perform these experiments (provided all other issues are carefully and fully dealt with), I do encourage you to develop your study as far as realistically possible for your next, revised version to strengthen your findings and increase their impact. If this is not possible, you will need to take the Reviewers' concerns and revise your conclusions. This Reviewer also notes that the appropriate and available models to test whether the changes observed indeed

protect from plaque rupture were not applied.

Reviewer 3 also proposes a number of items for your action.

In conclusion, while publication of the paper cannot be considered at this stage, we would be pleased to consider a suitably revised submission, provided that the Reviewers' concerns are fully addressed.

Please note that it is EMBO Molecular Medicine policy to allow a single round of revision only and that, therefore, acceptance or rejection of the manuscript will depend on the completeness of your responses included in the next, final version of the manuscript.

As you know, EMBO Molecular Medicine has a "scooping protection" policy, whereby similar findings that are published by others during review or revision are not a criterion for rejection. Although I clearly do not foresee such an instance in this case, I do ask you to get in touch with us after three months if you have not completed your revision, to update us on the status. Please also contact us as soon as possible if similar work is published elsewhere.

I look forward to receiving your revised manuscript as soon as possible.

***** Reviewer's comments *****

Referee #1 (Remarks):

Hoeksema et al provide a Report outlining the role of myeloid histone deacetylase-3 (Hdac3) in atherogenesis. LDL receptor knockout mice receiving LysM-Cre driven Hdac3 knockout bone marrow (Hdac3del) demonstrated a more stable plaque phenotype versus controls receiving Hdac3fl/fl marrow (Hdac3wt). Specifically, whilst overall plaque volume was increased, this appeared attributable to increased fibrous cap formation (rather than necrotic core volume, which decreased), and this was composed of collagen as opposed to increased vascular smooth muscle cells (VSMC). In vitro studies suggested that Hdac3del macrophage conditioned medium enhanced VSMC collagen production, implicating a secreted factor; microarray analyses went on to identify transforming growth factor-beta (TGF-beta) as a likely candidate. Additional in vitro studies confirmed increased TGF-beta secretion by Hdac3del macrophages, and the key role of this in VSMC collagen production; ChIP-seq analyses demonstrated Hdac3 binding near the Tgfb1 promoter, and enhanced acetylation of the Tgfb1 locus in Hdac3del macrophages. Further analysis of murine plaque characteristics suggested no difference in total macrophage area, but showed polarization toward an anti-inflammatory 'M2' type phenotype in Hdac3del mice, indicated by decreased iNOS, and increased Dectin-1, expressing cells. Plaque lipid content and macrophage size were also reduced in Hdac3del mice, and studies of peritoneal macrophages from these mice suggested reduced lipid accumulation, supporting the inference of reduced foam cell formation. Finally, analysis of human carotid atherosclerotic plaque samples suggested increased Hdac3 expression in vulnerable plaque, and positive correlation between Hdac3 and CD68 (macrophage marker), along with CD86 and HLA-DPB1 (activated macrophage markers) expression. Moreover, Hdac3 expression inversely correlated with that of Tgfb1.

These interesting and novel data provide a robust mechanistic link between myeloid Hdac3 activity and TGF-beta expression, along with their link to vulnerable atherosclerotic plaque characteristics. The manuscript is clearly written and the methods are appropriate. My suggestions to improve the manuscript are as follows:

- 1) Whilst it is important to see detailed atherosclerosis characterization in the aortic root, it is also important to consider the wider vasculature, which can show divergent phenotypes - can the authors provide data from the entire aorta?
- 2) An alternative model of Hdac3 silencing in grafted atherosclerotic vessels suggested a detrimental impact, potentially related to impaired endothelial cell survival (Circulation 2010; 121: 132-42). This suggests important tissue-specific differences in the potential therapeutic impact of Hdac3 inhibition. I note that conducting pharmacologic inhibitor studies (or global Hdac3 knockout) in mice to resolve this issue may be beyond the scope of this work if kept in the format of a Report, but

as a minimum these issues need to be discussed in the manuscript.

3) Assessment of atherosclerotic plaque stability/vulnerability is difficult using murine models. Therefore, statements such as 'confirming increased atherosclerotic plaque stability' (Results and Discussion, section one) should be reworded to be clear that such data can only be used to formulate inferences.

4) Can the authors provide any suggestions regarding what mediates increased Hdac3 activity in atherosclerotic plaque macrophages? This may highlight alternative therapeutic avenues.

5) It would be helpful to see cholesterol sub-fraction data (e.g. VLDL/LDL/HDL), instead of total cholesterol alone. The methods of triglyceride/cholesterol analysis should also be included in the experimental methodology.

6) Minor issues - please expand on BMM acronym at first use (I assume bone marrow macrophage); please add correct images to supplemental figure 3 as these appear inadvertently duplicated from supplemental figure 4.

Referee #2 (Comments on Novelty/Model System):

It has previously been reported that Hdac3 is essential for LPS-induced inflammatory gene expression and that Hda3 deletion promotes differentiation of alternatively activated macrophages. In these respects the data are not totally novel. However establishing a role for Hdac3 consistent with the paradigms these prior data supported to the setting of atherosclerosis is an important extension and has important implications for potential approaches to promoting plaque stability. Although the studies presented are well designed and controlled they are limited to the study of development of atherosclerosis in LDLR^{-/-} in a "prophylactic" setting in which the high fat diet required lesion is not introduced until after bone marrow chimerism is established. It would be very interesting to see if adoptive transfer of Hdac3^{-/-} monocytes/macros into LDLR^{-/-} with established lesions could also be impacted. In addition, the authors claim HDAC deletion in myeloid cells leads to stabilization of plaque - this conclusion needs to be tempered in that they only demonstrate a shift in lesion morphology believed to be consistent with plaque stability. They have not used the available models to test whether the changes observed indeed protect from plaque rupture. Such experiments would substantially increase the impact of the study but even if not required for publication, without such data the authors should be required to temper this conclusion. The extension to analysis of human tissues is a positive aspect of the study and even though correlative/descriptive, the human data increases the potential medical impact.

Referee #2 (Remarks):

The studies presented are well designed and controlled and the results of interest. However, the impact of the study would be enhanced if in addition to the analysis of myeloid deletion of Hdac3 in a preventive setting, their impact on established lesion were assessed. In addition, either the authors should use one of the now available models of plaque rupture to established that the impact on lesion morphology observed is indeed associated with plaque stability or if such studies are not provided then the authors should temper the conclusion by indicating the studies showed differences in lesion morphology consistent with the phenotype of more stable plaques rather than concluding they observed increased plaque stability.

Referee #3 (Remarks):

The authors transplanted Ldlr^{-/-} mice with bone marrow from wt or conditionally (lysM-Cre)-deleted HDAC3. They find slightly larger but much more stable lesions. The mechanism is increased TGF-beta expression by the HDAC3-deleted macrophages. The data are convincing and of high quality. Beyond the mouse work, the authors confirm correlations in vulnerable plaque from human endarterectomy specimens.

Minor concerns

1. The authors should show evidence of the degree of HDAC3 knockdown in BM and aortic macrophages. LysM-Cre is not always completely effective, especially in the BM.
2. Figure 1J is not explained. Why were these genes chosen? How was the activation index calculated?
3. The criteria for vulnerable plaques should be defined. The population used should be described. Why was HLA-DPB1 selected to stain for MHC-II? Is this allele expressed in all patients? How were the "paired stable controls" obtained?
4. Root lesions were measured and well defined. Aortic pinning and en face assessment could be added.

1st Revision - authors' response

02 June 2014

Referee #1 (Remarks):

1) Whilst it is important to see detailed atherosclerosis characterization in the aortic root, it is also important to consider the wider vasculature, which can show divergent phenotypes - can the authors provide data from the entire aorta?

Answer:

This is a good suggestion. However due to lack of material we were not able to assess atherosclerosis at a different location. We agree that another site may potentially show divergent phenotypes although this will depend on the stage of the development of the lesions at that particular site. However, since our characterizations in the aortic root are very well supported by our in vitro mechanistic studies, for the fibrotic phenotype and the inflammatory (M1/M2) phenotype, we feel that the outcome will be very much in line with the root data. In addition, we now have added additional mechanistic data supporting our findings on foam cell formation and lipid accumulation. Using ChIP-seq data we show that HDAC3 co-localizes with NCoR and PPAR/LXR peaks in the genome (new figures 2I and 2J and supplemental figure E5). It is known that HDAC3 in such way supports the break that the NCoR complex puts on nuclear receptors (Glass & Ogawa, 2006). Removal of HDAC3, leads to enhanced PPAR γ and LXR responses (as was already indicated by the original figure 1J showing induction of 'PPARG-ligand', 'PPARG' and 'LXR-ligand' pathways; and new figures 2G and 2H) leading to improved efflux of lipids from foam cells (new figure 2K).

2) An alternative model of Hdac3 silencing in grafted atherosclerotic vessels suggested a detrimental impact, potentially related to impaired endothelial cell survival (Circulation 2010; 121: 132-42). This suggests important tissue-specific differences in the potential therapeutic impact of Hdac3 inhibition. I note that conducting pharmacologic inhibitor studies (or global Hdac3 knockout) in mice to resolve this issue may be beyond the scope of this work if kept in the format of a Report, but as a minimum these issues need to be discussed in the manuscript.

Answer:

We fully agree that it is likely that cell- and tissue specific effects of Hdac3 silencing exist as indeed was illustrated by the endothelial Hdac3 data. We have added this point to the manuscript at page 5. Currently, no good HDAC3 specific pharmacological inhibitors are available so pharmacological inhibitor studies are not feasible right now. Moreover, total-body Hdac3 deficiency is lethal.

3) Assessment of atherosclerotic plaque stability/vulnerability is difficult using murine models. Therefore, statements such as 'confirming increased atherosclerotic plaque stability' (Results and Discussion, section one) should be reworded to be clear that such data can only be used to formulate inferences.

Answer:

We agree that plaque stability/vulnerability is difficult to apply to murine atherosclerotic lesions. Therefore, have carefully assessed our statements throughout the manuscript and reworded them where necessary.

4) Can the authors provide any suggestions regarding what mediates increased Hdac3 activity in atherosclerotic plaque macrophages? This may highlight alternative therapeutic avenues.

Answer:

This is a good point. It is likely that the most relevant activity regulation of HDAC3 lies in its interaction with other components of the NCoR repressor complex, which is inactivated upon nuclear receptor ligand-induced activation.

Transcriptional regulation of HDAC3 may be an additional layer of regulation. Not much is known about this. We have assessed HDAC3 expression under a wide variety of stimuli (see figure below) {figure removed upon request by author} but did not observe significant changes in expression. Although we cannot exclude that local atherosclerosis specific stimuli will lead to Hdac3 dysregulation, Hdac3 activity most likely is regulated at the post-transcriptional level, through modulation of its interaction with NCoR repressor complex members.

5) It would be helpful to see cholesterol sub-fraction data (e.g. VLDL/LDL/HDL), instead of total cholesterol alone. The methods of triglyceride/cholesterol analysis should also be included in the experimental methodology.

Answer:

We have performed cholesterol subfraction analysis. These experiments showed that the majority of the cholesterol is in the VLDL-sized fractions, but no differences were found between wild type and HDAC3 deficient groups. We have added these data as new supplemental figure E4D. We have also added the methods for lipid analysis.

6) Minor issues - please expand on BMM acronym at first use (I assume bone marrow macrophage); please add correct images to supplemental figure 3 as these appear inadvertently duplicated from supplemental figure 4.

Answer:

This has been added and corrected.

Referee #2 (Remarks):

The studies presented are well designed and controlled and the results of interest. However, the impact of the study would be enhanced if in addition to the analysis of myeloid deletion of Hdac3 in a preventive setting, their impact on established lesion were assessed. In addition, either the authors should use one of the now available models of plaque rupture to established that the impact on lesion morphology observed is indeed associated with plaque stability or if such studies are not provided then the authors should temper the conclusion by indicating the studies showed differences in lesion morphology consistent with the phenotype of more stable plaques rather than concluding

they observed increased plaque stability.

Answer:

We agree with the reviewer that an intervention study would be of great interest. One of the options may be to perform an adoptive transfer of monocytes/macrophages. However, our own experience has shown that adoptive transfer of monocytes/macrophages is not very effective in atherosclerosis. Although several papers have shown that adoptive transfer of lymphocytes can highly influence atherosclerosis development, transfer of monocytes/macrophages has minimal success. Very few cells migrate to the atherosclerotic lesions and it is very unlikely that these few cells influence plaque development significantly. Ideally, an intervention approach would be a pharmacological experiment. However, currently there are no selective Hdac3 inhibitors available preventing these studies.

We agree on the limitations with respect to plaque rupture and to our opinion it is difficult to mimic this in mice, despite some published potential models for rupture, they still not perfectly mimic rupture in human plaques. We thus have adapted the manuscript and throughout the paper we have carefully reworded phrases about plaque stability and rupture. We now refer to the effects that we see to a change towards “a more stable plaque phenotype”, which indeed best reflects the changes that we see.

To improve the impact of our study we have added additional mechanistic data explaining the differences in lipid loading upon HDAC3 deletion. Using ChIP-seq data we show that HDAC3 co-localizes with NCoR and PPAR/LXR peaks in the genome (new figures 2I and 2J and new supplemental figure E5). It is known that HDAC3 in such way supports the break that the NCoR complex puts on nuclear receptors (Glass & Ogawa, 2006). Removal of HDAC3, leads to enhanced PPAR α and LXR responses (as was already indicated by the original figure 1J showing induction of ‘PPAR α -ligand’, ‘PPAR α ’ and ‘LXR-ligand’ pathways; and new figures 2G and 2H) leading to improved efflux of lipids from foam cells (new figure 2K).

Referee #3 (Remarks):

The authors transplanted Ldlr^{-/-} mice with bone marrow from wt or conditionally (lysM-Cre)-deleted HDAC3. They find slightly larger but much more stable lesions. The mechanism is increased TGF- β expression by the HDAC3-deleted macrophages. The data are convincing and of high quality. Beyond the mouse work, the authors confirm correlations in vulnerable plaque from human endarterectomy specimens.

Minor concerns

1. The authors should show evidence of the degree of HDAC3 knockdown in BM and aortic macrophages. LysM-Cre is not always completely effective, especially in the BM.

Answer:

Indeed this is an important issue. We have quantified HDAC3 expression in both BMM and peritoneal macrophages (PM) (new figure 2G and new supplemental figure E2C). Deletion is 60-75%, which is relatively good for the LysMCre system. We were not able to quantify deletion in the plaque macrophages, but because the atherosclerosis phenotype (fibrosis, inflammation, lipids) is so well reflecting our in vitro BMM and PM data, we are confident that there is a substantial degree of deletion in plaque macrophages as well.

2. Figure 1J is not explained. Why were these genes chosen? How was the activation index

calculated?

Answer:

These gene lists were generated by the Upstream Regulator analysis of the Ingenuity pathway analysis software based on the whole micro-array datasets. In the Upstream regulator analysis which we performed, the upstream regulators that are shown are the most significant predicted upstream regulators. We ranked on z-score, which is a statistical test used to define the overlap between differentially expressed genes and genes that are known to be affected by the upstream regulator.

3. The criteria for vulnerable plaques should be defined. The population used should be described. Why was HLA-DPB1 selected to stain for MHC-II? Is this allele expressed in all patients? How were the "paired stable controls" obtained?

Answer:

Plaque progression stage was assessed by examination of Hematoxylin-Eosin stained slides from flanking segments using the Virmani classification criteria by a cardiovascular pathologist and an experienced researcher in cardiovascular pathology. Segments designated as stable featured either a fibrous cap atheroma or pathological intimal thickening. Segments designated as ruptured included a thrombus and/or presented intraplaque haemorrhage. Sections were considered stable (S) or ruptured (R) if they were flanked at both sides within the same endarterectomy specimen, by segments of identical progression stage. Only endarterectomy specimen were used, which contained both stable and ruptured plaque segments. In the revised manuscript, we now provide a more detailed and unambiguous description of the classification criteria and replaced the confusing term "vulnerable" by the more correct "ruptured".

HLA-DPB1 was not used to stain for MHC-II but was used to correlate the mRNA expression of HDAC3 to the mRNA expression of HLA-DPB1, which is, besides a standard HLA molecule, also a marker of macrophage activation.

The HLA-DPB1 gene, encoding the DP beta chain of class II molecules, was not expressed in two samples which were left out of that particular analysis.

Atherosclerotic plaque samples obtained during carotid endarterectomy (CEA) were collected from the Maastricht Pathology Tissue Collection (MPTC). The endarterectomy specimens were cut into parallel, transverse segments of 2-3 mm thickness and fixed. Hematoxylin-Eosin stained slides from these flanking segments were classified as described above. For the present study analysis was performed on samples that were flanked by two segments of identical classification, be it stable or ruptured; and were derived from CEA specimen that contained plaque segments of both classifications. We included in our analysis 20 such pairs (stable-ruptured) from independent patients.

4. Root lesions were measured and well defined. Aortic pinning and en face assessment could be added.

We have not been able to perform these analyses because of current lack of these materials. However, as said above our in vivo aortic root data is so well reflecting our in vitro mechanistic studies that we are confident that atherosclerosis at other locations will have a similar phenotype, of course depending on the degree of atherosclerosis.

I am following up on the decision letter sent to you a few minutes ago (see below). I just realised that there are a number of mistakes in the Author Contributions section (incomplete acronyms such as for Boshuizen, Biessen and Daemen, missing one for Dallinga-Thie and finally, unclear meaning of G.M.D.).

Please take care of this too in your revision.

Thank you.

June 5, 2014

Thank you for the submission of your revised manuscript to EMBO Molecular Medicine. We have now received the enclosed reports from the Reviewers that were asked to re-assess it. As you will see the reviewers are now supportive (but please take action on Reviewer 1's comment) and I am pleased to inform you that we will be able to accept your manuscript pending the following final amendments:

1) As per our Author Guidelines, the description of all reported data that includes statistical testing must state the name of the statistical test used to generate error bars and P values, the number (n) of independent experiments underlying each data point (not replicate measures of one sample), and the actual P value for each test (not merely 'significant' or ' $P < 0.05$ ').

2) Every published paper now includes a 'Synopsis' to further enhance discoverability. Synopses are displayed on the journal webpage and are freely accessible to all readers. They include a short standfirst - to be written by the editor - as well as 2-5 one-sentence bullet points that summarise the paper (to be written by the author). Please provide the short list of bullet points that summarise the key NEW findings. The bullet points should be designed to be complementary to the abstract - i.e. not repeat the same text. We encourage inclusion of key acronyms and quantitative information. Please use the passive voice. Please attach these in a separate file or send them by email, we will incorporate them accordingly.

3) We are now encouraging the publication of source data, particularly for electrophoretic gels and blots, with the aim of making primary data more accessible and transparent to the reader. Would you be willing to provide a PDF file per figure that contains the original, uncropped and unprocessed scans of all or at least the key gels used in the manuscript? The PDF files should be labeled with the appropriate figure/panel number, and should have molecular weight markers; further annotation may be useful but is not essential. The PDF files will be published online with the article as supplementary "Source Data" files. If you have any questions regarding this just contact me.

I also thought that you might like to know that I have commissioned an expert to write a Closeup (our version of a News and Views article) to highlight your work, and which will be published alongside your article in an upcoming EMBO Molecular Medicine issue.

I look forward to receiving the revised manuscript, files and information as soon as possible, possibly no later than two weeks from now.

***** Reviewer's comments *****

Referee #1 (Remarks):

I note the rebuttal letter, and the accompanying amendments to the manuscript, which have appropriately addressed most of my comments. Whilst it is unfortunate that data are not available to describe atherosclerosis in the entire aorta, I do not feel that this is absolutely essential. The additional mechanistic data appear valid and provide further support for the authors' conclusions.

Minor issue - a word appears missing between 'we' and 'several' on page 4 in the sentence 'Likewise, in the absence of Hdac3 we several PPAR and LXR target genes to be upregulated in macrophages (Fig 2G-H)'.

Referee #3 (Comments on Novelty/Model System):

adequately revised

Referee #3 (Remarks):

adequately revised

2nd Revision - authors' response

06 June 2014

We thank you and the reviewers for the quick handling and the review of our manuscript and adapted the last issues.

We corrected the error in the sentence that was mentioned by reviewer 1 and the mistakes in the acronyms in the Author Contribution section. Furthermore, we added the actual P value for each test in the Figure legends and attached a document with bullet points for the synopsis.